# Increased Local Testosterone Levels Alter Human Fallopian Tube mRNA Profile and Signaling

**DOI:** 10.3390/cancers15072062

**Published:** 2023-03-30

**Authors:** Angela Russo, Brian P. Cain, Tia Jackson-Bey, Alfredo Lopez Carrero, Jane Miglo, Shannon MacLaughlan, Brett C. Isenberg, Jonathan Coppeta, Joanna E. Burdette

**Affiliations:** 1Department of Pharmaceutical Sciences, University of Illinois Chicago, Chicago, IL 60607, USA; 2Charles Stark Draper Laboratory, Cambridge, MA 02139, USA; 3Department of Obstetrics and Gynecology, University of Illinois Chicago, Chicago, IL 60607, USA

**Keywords:** fallopian tube, ovarian cancer, testosterone, WNT4, microfluidics

## Abstract

**Simple Summary:**

Increased testosterone has been associated with increased risk of ovarian cancer. High Grade Serous Ovarian Carcinoma (HGSOC) mostly originates from the fallopian tube epithelium (FTE), however, the stepwise events that occur during its tumorigenic transformation are unknown. Early lesion models of fallopian tube cancer stimulate the ovary to produce more androgen, supporting the hypothesis that high levels of testosterone may contribute to HGSOC early development. Nevertheless, the role of androgens in ovarian cancer has not been investigated. This study addresses the role of androgen in the early development of ovarian cancer from primary human FTE. Our results show that increased testosterone alters mRNA profile of hFTE, increasing WNT4 and LGR6 expression as well as increasing the migratory ability of fallopian tube epithelial cells.

**Abstract:**

Fallopian tube epithelium (FTE) plays a critical role in reproduction and can be the site where High Grade Serous Ovarian Carcinoma (HGSOC) originates. Tumorigenic oviductal cells, which are the murine equivalent of human fallopian tube secretory epithelial cells (FTSEC), enhance testosterone secretion by the ovary when co-cultured with the ovary, suggesting that testosterone is part of the signaling axis between the ovary and FTSEC. Furthermore, testosterone promotes proliferation of oviductal cells. Oral contraceptives, tubal ligation, and salpingectomy, which are all protective against developing ovarian cancer, also decrease circulating levels of androgen. In the current study, we investigated the effect of increased testosterone on FTE and found that testosterone upregulates wingless-type MMTV integration family, member 4 (WNT4) and induces migration and invasion of immortalized human fallopian tube cells. We profiled primary human fallopian tissues grown in the microfluidic system SOLO-microfluidic platform –(MFP) by RNA sequencing and found that p53 and its downstream target genes, such as paired box gene 2 (PAX2), cyclin-dependent kinase inhibitor 1A (CDK1A or p21), and cluster of differentiation 82 (CD82 or KAI1) were downregulated in response to testosterone treatment. A microfluidic platform, the PREDICT-Multi Organ System (PREDICT-MOS) was engineered to support insert technology that allowed for the study of cancer cell migration and invasion through Matrigel. Using this system, we found that testosterone enhanced FTE migration and invasion, which was reversed by the androgen receptor (AR) antagonist, bicalutamide. Testosterone also enhanced FTSEC adhesion to the ovarian stroma using murine ovaries. Overall, these results indicate that primary human fallopian tube tissue and immortalized FTSEC respond to testosterone to shift expression of genes that regulate invasion, while leveraging a new strategy to study migration in the presence of dynamic fluid flow.

## 1. Introduction

Increased serum androgen levels are associated with ovarian cancer [1] and endometrial cancer [2]. Testosterone represses cilia function in the fallopian tube [3], and promotes proliferation of murine oviductal cells, which are the equivalent of fallopian tube epithelium (FTE) [4]. High Grade Serous Ovarian Carcinoma (HGSOC) can originate from the FTE cells [5], but the role of androgens in FTE function has only been evaluated in a few studies [3,4,6,7]. Previously, testosterone was shown to be enhanced after co-culture of the ovary in proximity to tumorigenic fallopian tube cells and increased oviductal cell proliferation and invasion, suggesting that androgen might alter FTE signaling [4]. Oral contraceptives, which reduce the risk of developing ovarian cancer, decrease circulating levels of androgens [8,9]. Lastly, tubal ligation has been shown to marginally reduce the testosterone precursor dehydroepiandrosterone (DHEA) [10,11] which is also associated with increased risk of ovarian cancer [12].

Women with polycystic ovary syndrome (PCOS) often present with increased androgen and ovulatory dysfunction-associated infertility. While most factors that reduce the lifetime number of ovulations are protective against developing ovarian cancer, women with PCOS do not have reduced risk and some studies even show an increased risk [1,13,14]. Androgen regulates mRNA levels and the activity of p53 [15,16,17] through four androgen-responsive elements on the *TP53* gene promoter [17]. *TP53* is mutated in 96% of HGSOC [18,19] and the loss or mutation of *TP53* is considered an early event in FTE carcinogenesis [20]. Wild type *TP53* negatively regulates the expression of the androgen receptor (AR) in prostate cancer cell lines [19] indicating that both AR and p53 regulate each other, and that intact tumor suppressor function of p53 tends to inhibit androgen signaling whereas loss of *TP53* enhances AR signaling. Transformation of human fallopian tube cells into secretory cell outgrowth (SCOUT) is often described as a loss of PAX2 and an expansion of sixteen or more secretory cells, which occurs along with a reduction in ciliated cells. Mutation and loss of *TP53* prevents cilio genesis [21,22], and testosterone represses p53 and cilia gene expression.

In the present study, we investigated the role of testosterone on human primary fallopian tube tissues as well as on immortalized secretory cell lines. We cultured the human fallopian tube epithelial tissues using organ-on-a-chip microfluidic technology called PREDICT-multi-organ systems (PREDICT-MOS). Advancing the capabilities of our previous single organ platform PREDICT96 [23], PREDICT-MOS allows the co-culture of *n* = 30 tissue pairs, while allowing optimal exchange of nutrients in either a submerged or an air-liquid interface tissue format. In this study we showed that addition of exogenous testosterone reduced apoptotic signals, such as p21, while increasing cancer stem cell markers WNT4 and LGR6. We also showed that increased testosterone induced migratory and invasive properties of human FTE cells. Together, our data suggest that increased testosterone can induce changes of the fallopian tube epithelial cells, possibly increasing early events leading to ovarian cancer.

## 2. Materials and Methods

### 2.1. Cell Culture

FT33-TAg and FT190 human fallopian tube secretory epithelial cell lines (FTSEC), immortalized with hTERT and SV40, were provided by Ronny Drapkin [24,25] and were cultured as previously described [26,27].

FTE33-TAg cells and MOE cells were grown in the MEMα with 10% FBS (Gibco, Baltimore, MD, USA), L-glutamine (2 mmol/L, Gibco, Baltimore, MD, USA), EGF (0.1 mg/mL, Roche, Basel, Switzerland), ITS (Roche, Basel, Switzerland), gentamicin (50 mg/mL, Gibco), 17β-estradiol (1 mg/mL in 100% EtOH, Sigma Aldrich, St. Louis, MO, USA) and penicillin/streptomycin. All of the cells were passaged a maximum of 20 times and cultured in the monolayers in 95% air and 5% CO_2_ at 37 °C in a cell incubator (Sanyo, Osaka, Japan) according to the ATCC cell culture protocol. The shRNA lines’ media contained puromycin for maintenance.

### 2.2. Human Fallopian Tissue

Human fallopian tube tissues were provided by the Department of Obstetrics and Gynecology at the University of Illinois (Chicago, IL, USA). Written informed consent was obtained from each subject prior to surgery. The tissue was dissected in MEM and Nutrient Mixture F-12 medium (50:50) with 10% fetal bovine serum within a 24-h period from arrival and then transferred to MEM with 0.3% bovine serum albumin, 0.5 mg/mL fetuin, 1% penicillin/streptomycin and ITS medium. The tissue received was cut lengthwise and the inner epithelium was then isolated. Epithelium from the same patient was divided into normal and PCOS-like conditions and cultured on 0.4 µm pore Millicell inserts (Millipore, MA, USA). After 24 h, the static condition experiments continued in static culture in 12-well plates while microfluidic experiments were transferred to the microfluidic devices for dynamic culture. Both static and microfluidic conditions were maintained in incubators at 37 °C and 5% CO_2_. The media condition, referred to as “low,” contained estrogen (E_2_) 0.1 nM and testosterone 0.8 nM on Days 0–6 then E_2_ 1 nM and testosterone 0.8 nM on Days 7–13 and finally E_2_ 1 nM and testosterone 1.25 nM for 1 day prior to the end of the experiment on Day 14. The media condition, referred to as “high,” contained contained E_2_ 0.1 nM and testosterone 2 nM on Days 0–6 then E_2_ 1 nM and testosterone 2 nM on Days 7–14 until the experiment ended on Day 14. In this model, recombinant follicle-stimulating hormone (rFSH) was present in the media from Days 0 to 14, to replicate the follicular phase of the menstrual cycle. After 14 days of culture, the tissue was either frozen for RNA extraction or fixed in 4% paraformaldehyde for immunohistochemistry.

### 2.3. RNA Isolation, cDNA Synthesis and RT-PCR

RNA extraction was performed as previously described [28]. iScript^TM^ cDNA synthesis kit (Bio-Rad, Hercules, CA, USA) and SYBR green (Roche, Madison, WI, USA) were used according to manufacturer’s instructions. All qPCR runs were performed on the CFX96 (Bio-Rad, San Francisco, CA, USA). Primers used are described in Appendix A**.**

### 2.4. Spheroid Formation

Cells were detached with trypsin/EDTA, collected with media, counted, and diluted to 2000 cells/mL. Five hundred cells were added to each well of a 96-well, round-bottom set of ULA plates (Corning, NY, USA). Cells were incubated for 10 days. The contents of each well were imaged (undisturbed), pipetted up and down with a 200 μL pipette and imaged again (disturbed) with 4× or 10× objectives. The diameter of the spheroids was measured via ImageJ (National Institute of Health, Washington, DC, USA). Experiments were performed on 2–3 wells replicates on each plate. Experiments were replicated independently at least three separate plates.

### 2.5. Wound Healing Assay

Cells were seeded at 5 × 10^4^ cells per well. After 24 h, a scratch was performed and the medium was removed and substituted with medium containing vehicle or hormones. Pictures were taken right after the scratch and 24 h later using an AmScope MU900 with Toupview software (AmScope, Irvine, CA, USA). The scratch areas were measured using ImageJ software.

### 2.6. Boyden Chamber Invasion Assay

Matrigel was thawed out on ice and diluted to 300 μg/mL with serum-free medium. 120 μL of diluted Matrigel was added to the 0.8 μm tissue scaffold of the Boyden chamber and incubated at 37 °C for 1 h for Matrigel to solidify. Excess of Matrigel was removed from tissue scaffold. Five hundred microliters of complete medium (containing 10% FBS) was added to 24 well plates at the bottom of the tissue scaffold. Cells were trypsinized and collected in media with FBS. Cells were washed 2× with serum-free MEM and resuspended in serum-free medium. 120 μL of FTSEC cells at 1 × 10^4^ cells/well was added and incubated for 24 h. Cells from the top of the tissue scaffold were removed with a cotton swab. Tissue scaffolds fixed with 4% PFA for 5 min, permeabilized with 70% methanol for 5 min, stained with 0.2% crystal violet in 10% ethanol for 10 min. Tissue scaffolds were then rinsed 2× with PBS and dried overnight. Images of each tissue scaffold were taken using AmScope MU900 with Toupview software (AmScope, Irvine, CA, USA) and invading cells (at the bottom of the tissue scaffold) were counted in ImageJ.

### 2.7. Dynamic Invasion in the PREDICT-MOS Microfluidic System

The tissue scaffolds were washed with 70% ethanol and dried in the hood. Then the tissue scaffolds were disposed on a flat sterile surface and coated with 120 μL Matrigel for 40 min at room temperature. The excess of Matrigel was removed and the tissue scaffold was placed in the aluPlate. 25 × 10^4^ cells labeled with CellTracker (Thermo Fisher Scientific, Waltham, MA, USA) or unlabeled were added to each tissue scaffold. Media with testosterone, testosterone plus 10 μM Bicalutamide or vehicle DMSO were added in the donor well. After 24 h the non-migrating cells were removed with a cotton swab and cells at the bottom were stained and imaged as for Boyden chamber assay. Labeled cells were counted using AmScope MU900 with Toupview software (AmScope, Irvine, CA, USA) and invading cells (at the bottom of the tissue scaffold) were counted in ImageJ.

### 2.8. Ex Vivo Colonization Assay

Ovaries were removed from day 16- or 17-day old CD1 mice and were cut with a scalpel blade to expose ovarian stroma as described earlier [29,30]. Each ovary was incubated with 30,000 fluorescently labeled cells with and with synthetic androgen R1881 or vehicle overnight at 37 °C in an orbital shaker (40 rpm). The next day ovaries were washed several times to remove non-attached cells, all observable cells were counted, and representative pictures were taken with an AmScope MU900 with Toupview software (AmScope, Irvine, CA, USA).

### 2.9. Immunohistochemistry (IHC)

Reproductive tracts Tissues were fixed in 4% PFA, embedded in paraffin, processed, and prepared for immunohistochemistry as previously reported [26,31,32]. Tissues were incubated with primary antibodies overnight (Appendix A). Images were acquired on a Nikon Eclipse E600 microscope using a DS-Ri1 digital camera and NIS Elements software.

### 2.10. Statistical Analyses

All data are represented as mean ± standard error. Statistical analysis was carried out using Prism software (GraphPad, La Jolla, CA, USA). All conditions were tested in three replicates in at least triplicate experiments. Statistical significance was determined by Student’s t-test when only two populations were compared; one-way ANOVA when more than two population were compared; two-way ANOVA when more than two populations were each divided into groups. * *p* < 0.05, ** *p* < 0.01, *** *p* < 0.001, **** *p* < 0.0001 were considered significant; ns indicates non-significant.

## 3. Results

### 3.1. RNAseq Analysis Revealed That High Testosterone Regulates Genes Involved in Early Tumorigenesis

Previously, we found that tumorigenic oviductal cells grown in proximity to the ovary enhance testosterone, which may provide a unique signal in the tumor microenvironment to increase oviductal cells proliferation and invasion [4]. Further, we demonstrated that testosterone could repress cilia gene expression in human fallopian tube primary tissues [3], which may contribute to the outgrowth of secretory cells, a putative early event in fallopian tube transformation. RNAseq [3] was conducted on human primary tissue grown in a Solo-MFP microfluidic platform and revealed upregulation of migratory pathways and downregulation of apoptotic pathways (Appendix A). The heat map, generated from these data, shows the top 25 genes upregulated and downregulated by testosterone (Figure 1A) with testosterone at high (2 nM) and low (physiological-0.8 nM) levels based on serum concentrations found in PCOS women as compared to controls [33,34,35] (Figure 1B). We validated those shown in the literature to be specific to androgen stimulation [3] such as STEAP4v1, STEAP4v3, ZBTB16, and KIF5C (Figure 1C).

Testosterone decreases TP53 mRNA and target genes in the fallopian tube. Pathways regulated by increased testosterone levels using the David Bioinformatic database (KEGG) showed regulation of cancer-related pathways shown in the heat map in Figure 2A.

Further, high expression of AR is associated with lower overall survival (Figure 2B). To validate the RNAseq demonstrating regulation of tumorigenic transcripts, we performed qPCR on tissue grown in both the Solo-MFP microfluidic as well as in static conditions. Regulation of these target genes was conserved in both the static and dynamic culture conditions (using Solo-MFP) in response to testosterone (Figure 3A,B). We validated downregulation of TP53, p21, PAX2, CD82, and laminin subunit alpha 2 (LAMA2), upregulation. PAX2 is reduced at the mRNA level when tp53 is mutated in the DNA binding domain or when wild-type p53 is reduced [36]. LAMA2, which regulates migration, was upregulated and is known to be upregulated by mutant TP53 [37]. We also found a reduction of the p53 downstream genes p21 and CD82, which regulate the cell cycle and act as a tumor suppressor, respectively.

### 3.2. High Testosterone Induces mRNA Expression of Cancer Stem Cell Markers

In the literature, AR regulates ovarian cancer stem cells by regulating Nanog that is a regulator of cancer stem cells pluripotency and self-renewal [38,39]. We found that testosterone repressed ciliated gene expression and therefore may reduce differentiation in the fallopian tube [3]. WNT4 and LGR6 are stem-cell markers found in the fimbriae of the tube [40,41]. We validated expression changes of stem-like markers found in the RNAseq using qPCR and immunohistochemistry. While our initial RNAseq was collected from culturing in Solo-MFP (*n* = 4), here we performed studies in the higher replicate PREDICT-MOS and optimized it for the growth of human primary fallopian tube tissues [42]. We found that high testosterone (2 nM) as compared to low (0.8 nM) upregulated WNT4 and LGR6 mRNA levels in primary tissue cultured in PREDICT-MOS as well as in static culture (Figure 4A,B). Next, we treated human immortalized FTE cell lines, FT33-TAg and FT190, in static conditions and found that testosterone enhanced expression of LGR6 and WNT4 (Figure 4C,D). We then used IHC to determine whether WNT4 and LGR6 were upregulated in the primary tissues and found enhanced protein staining of both in response to testosterone treatment in the dynamic culture (Figure 4E). We also tested whether high testosterone increased the ability of FTSEC cells to form spheroids and our results show that high testosterone did not increase the size of spheroids grown in ultra-low attachment conditions (Appendix A).

### 3.3. High Testosterone Increased the Migratory and Invasive Ability of Human Fallopian Tube Cells through AR in a Microfluidic Platform

While our initial RNAseq was collected from culturing in Solo-MFP, we modified a higher throughput system called PREDICT96 into what we termed the PREDICT-Multi-Organ System (MOS) (Appendix A), which is optimized to grow human primary fallopian tube tissues with sufficient material for interrogating protein and RNA changes using an air-liquid interface culture of dissected primary tissues [42]. The advantages of the MOS include the ability to perform multi-organ interactions. The SOLO-MFP is a single organ system. Therefore, the MOS has a higher throughput with *n* = 30 single organs or *n* = 15 interacting organs (Appendix A).

Previously we found that murine oviductal cells invaded through Matrigel in a Boyden chamber in the presence of androgen [4]. In order to extend these findings into human cell lines, the human immortalized cells FT33-TAg were treated with low and high testosterone for 72 h and migration was assessed by wound healing assay. Increased testosterone significantly increased the wound closure after 24 h (Figure 5A). FT33-TAg were treated with low and high testosterone for 72 h and invasion was assessed by Boyden chamber invasion assay. Increased testosterone increased the invasive ability of the cells over 24 h (Figure 5B). We then developed an insert technology that held an 8-micron pore transmembrane that fits the PREDICT-MOS wells such that cell lines can be grown on this membrane and invasion can be detected in response to testosterone (Figure 6A). To determine if testosterone induced invasion in these dynamic conditions, cells were plated on the 8-micron PREDICT-MOS tissue transmembrane, which had been previously coated with Matrigel. Media with low or high testosterone concentrations were added to the donor well and pumped through the system. We found that testosterone increased invasion after 48 h treatment in FT33-TAg and FT190 fallopian tube lines (Figure 6B,C). To confirm that testosterone-induced migration was dependent on the nuclear action of the AR, cells were plated on the 8-micron PREDICT-MOS tissue scaffold, as described above. Media low testosterone, high testosterone or high testosterone plus the AR antagonist, bicalutamide, were added to the donor well. Bicalutamide blocked testosterone increased invasion in FT33-TAg and FT190 fallopian tube lines (Figure 6D,E).

### 3.4. High Testosterone Increased the Ability of Human Fallopian Tube Cells to Attach to Ovarian Stroma

We have shown previously that the loss of PTEN increased the production of WNT4 in the FTE cells and promoted the attachment of oviductal cells to ovarian stroma [29]. Therefore, we tested if testosterone could increase attachment to the ovarian stroma particularly since it also increased WNT4 in the human tissue and immortalized cell lines. The experiment was performed in static conditions as previously described [29] since a dynamic protocol for this assay was not yet optimized. Our results also show that testosterone increased FT33-Tag attachment to the ovary, but not FT190 cells (Figure 7A–C).

## 4. Discussion

The fallopian tube epithelium (FTE) is the site of origin of High-Grade Serous Ovarian Carcinoma (HGSOC). The mechanisms leading to early tumorigenesis from FTE are not fully understood, however the process of ovulation increases the risk for developing the disease [43]. Our lab has also shown that ovarian-secreted factors can induce FTE tumorigenic signaling such as enhancing ovarian attachment and altering expression of important pathways [42]. Oral contraceptives, lactation, and salpingectomy reduce the risk of ovarian cancer and also reduce the total lifetime number of ovulations [43]. However, women with PCOS, despite anovulation, are not protected from ovarian cancer risk, which may be due to hyperandrogenism that is a hallmark of PCOS. In super-ovulated mice, increased expression of AR was detected in the FTE [44] suggesting that ovulatory factors may render FTE more sensitive to testosterone. Androgen treatment increased oviductal cell lines proliferation, however its role on human fallopian tube function has not extensively studied. Stimulation of human fallopian tube primary tissues with 2 nM testosterone as compared to 0.8 nM caused a significant reduction in mRNA encoding for cilia machinery and slowed cilia beat frequency [3] suggesting that testosterone might hinder cilia cell health and therefore potentially facilitate secretory cell outgrowth. Testosterone was also shown to promote proliferation of murine oviductal cells which are the equivalent of FTE cells [4]. Recently, testosterone secretion from the ovary has been shown to be augmented by co-culturing the ovary in proximity to tumorigenic FTE cells [4], suggesting that increased testosterone may favor tumor cell survival and proliferation. Most procedures protecting against ovarian cancer, such as oral contraceptives [8,9], tubal ligation, and salpingectomy [10,11,45] also decrease circulating levels of androgens. Nevertheless, the mechanism of androgen action on HGSOC tumorigenesis has not been determined.

Herein, we have also shown that testosterone increases migration and invasion of human immortalized fallopian tube cells, and it was dependent on AR signaling since bicalutamide, an androgen receptor antagonist, reduced testosterone-induced invasion. We also assessed the ability of testosterone to induce invasion in a microfluidic system, thereby addressing whether invasion can occur in dynamic conditions where the flow of nutrients is continuously pumped to the cells. The optimization of a dynamic invasion will allow monitoring real-time invasion induced by hormonal changes occurring during the women cycle. High testosterone increased the expression of WNT4 and LGR6, which have both been described as cancer stem cell (CSC) markers in both human tissues and immortalized cell lines. A previous study showed that androgen can regulate CSCs function through regulation of cancer stem cell markers [39]. Increased testosterone also reduced p53 and p21 mRNA levels as well as PAX2 levels. Loss of PAX2 expression and loss of normal p53 function have been reported in early benign lesions of the fallopian tube and have been shown to increase CSC functions [46,47,48]. DHT and p53 have been reported to regulate each other in prostate function and pathology, however, there is no report of the role of testosterone in regulating p53 in fallopian tube.

Previous animal models where either DHT or DHEA was given chronically to female rodents have reported increased proliferation of ovarian surface epithelium without inducing tumor formation [49], therefore, testosterone has not been shown to induce FTSEC tumorigenesis in vivo but changes in early events leading to precursor lesions have not been studied. Another study showed that androgen induces proliferation of murine oviductal cells [4]. We used testosterone, at concentrations relevant in human physio-pathology, instead of DHT or DHEA. Indeed, our low concentration of testosterone is based on serum concentrations reported in women, whereas the high is 2-fold higher and is consistent with testosterone serum levels in polycystic ovarian syndrome (PCOS) patients [33,34,35]. We used synthetic androgen, R1881 in our co-culture study to test the adhesion of fallopian tube cells to the ovary since the ovary expresses aromatase and could metabolize testosterone to estrogen. Bicalutamide, an anti-androgen, was not consistently successful in treating ovarian cancer in early clinical trials, although second generation enzalutamide is now being evaluated in phase II clinical trials [50,51]. The effects of bicalutamide on reducing ovarian cancer cell proliferation are less significant after treatment with chemotherapeutic due to reduced levels of AR [49]; therefore, the use of anti-androgen should be considered based on expression of AR and may be more impactful prior to other chemotherapy or as cancer prevention. Although HGSOC is typically diagnosed after menopause, its development starts several years earlier. Testosterone decreases with age, but it increases in postmenopausal women compared to premenopausal women [1,52]. Additionally, the reduction is much lower than estrogen leaving an unbalanced ratio of estrogen/testosterone levels that could be even more pronounced locally at the ovary-FTE interface. This question could be investigated using a microfluidic system that allows studying ovulation in real time as previously described [53,54]. In summary, our results are the first to show how local stimulation of FTE with testosterone can affect the function of these cells and render them more migratory and invasive.

## 5. Conclusions

Testosterone plays a role in women’s reproduction and its level can rise due to pathological condition such as PCOS leading to increased risk of ovarian cancer. High grade serous ovarian cancer mainly originates from the fallopian tube epithelium. Herein, we show that testosterone can exert a function on fallopian tube epithelium inducing changes in gene expression such as increase in LGR6 and WNT4. In addition, we show that increased testosterone favor migration and invasion of fallopian tube secretory epithelial cells as well as adhesion to the ovarian stroma. Taken together, these experiments suggest that increased testosterone levels may contribute to an enhancement of the migratory properties of the fallopian tube epithelium.

## Figures and Tables

**Figure 1 cancers-15-02062-f001:**
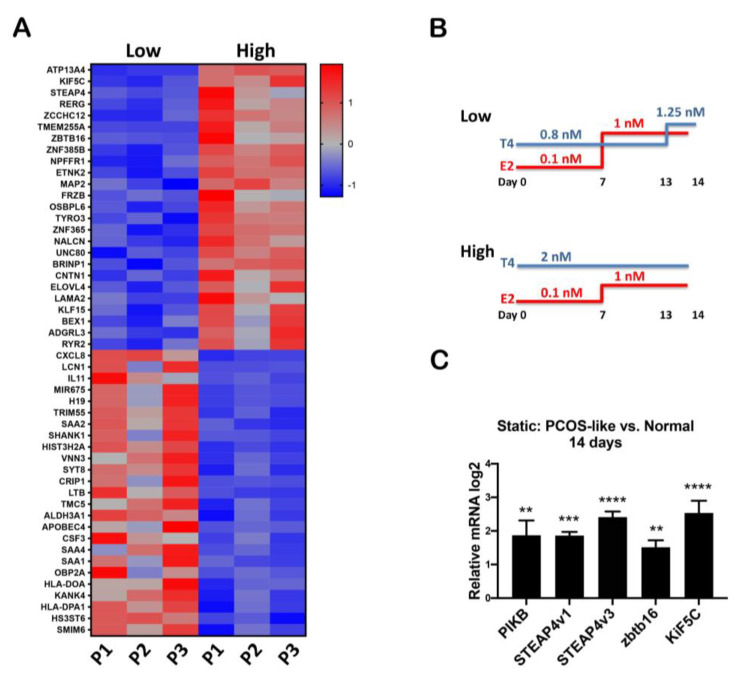
**Increased testosterone alters the primary human fallopian tube epithelium gene profile.** (**A**) Heat map from RNAseq from primary human fallopian tube of three different patients treated with low (0.8 nM) and high (2 nM) testosterone for 14 days in Solo-MFP. (**B**) Protocol of testosterone concentration used to stimulate hFTE. (**C**) qPCR validating genes regulated by testosterone in RNAseq from the same patients’ samples used for RNAseq. One-way Anova was performed and ** *p* < 0.01, *** *p* < 0.001, **** *p* < 0.0001 were considered significant.

**Figure 2 cancers-15-02062-f002:**
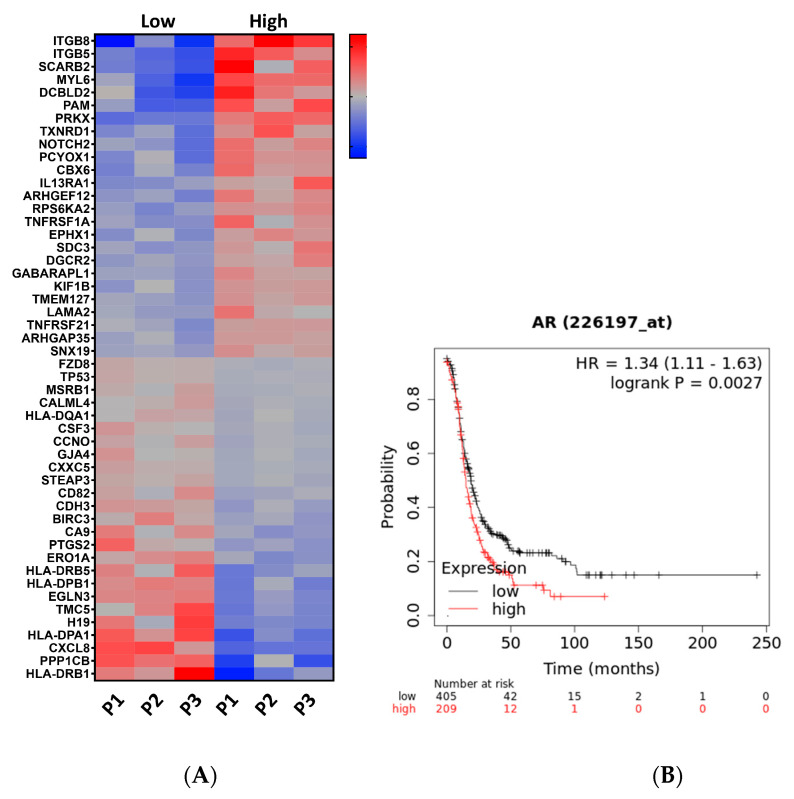
**Increased testosterone signaling in fallopian-tube-derived ovarian cancer genesis.** (**A**) Heat-map showing tumorigenic genes regulated testosterone low (0.8 nM) and high (2 nM) in 3 different human primary tissues in microfluidic conditions. (**B**) Overall survival for high versus low AR in serous ovarian cancer (https://kmplot.com/analysis/index.php?p=service&cancer=ovar, accessed in January 2023).

**Figure 3 cancers-15-02062-f003:**
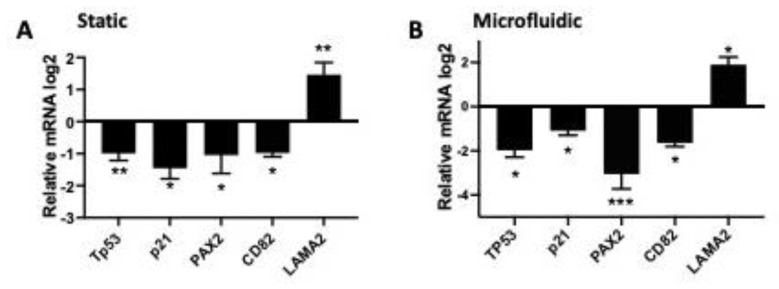
**High testosterone reduces TP53 function.** qPCR analysis of 3 independent experiments in human FTE tissues in static (**A**) and dynamic (**B**) conditions in the SOLO-MFP. One-way Anovawas performed and * *p* < 0.05, ** *p* < 0.01, *** *p* < 0.001 were considered significant.

**Figure 4 cancers-15-02062-f004:**
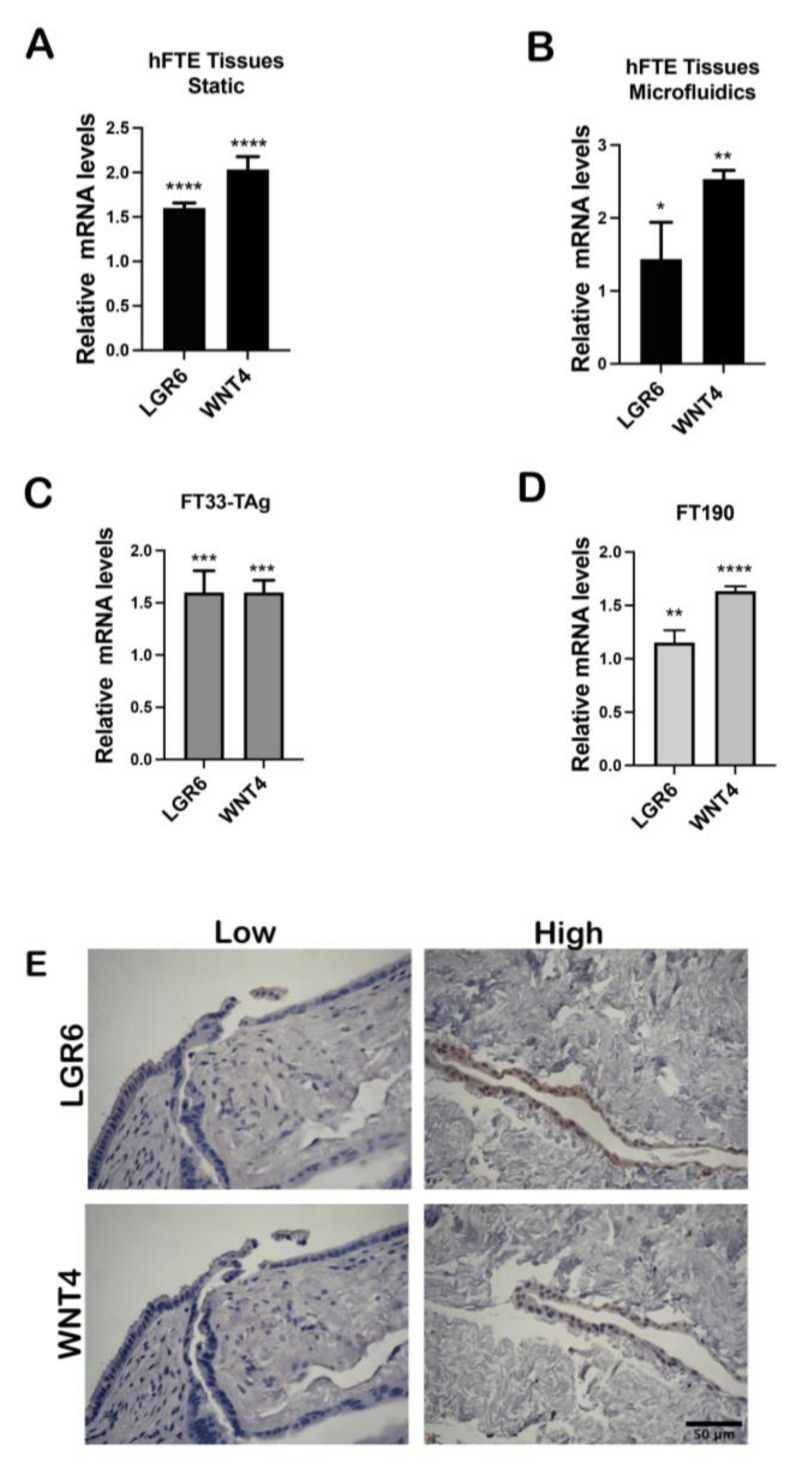
**Testosterone increases mRNA expression of cancer stem cells markers.** (**A**,**B**) qPCR analysis of cancer stem cells markers WNT4 and LGR6 in primary fallopian tube epithelium tissue grown in static and microfluidics from the same patients as RNAseq data or (**C**,**D**) immortalized FTE FT33-TAg and FT190 grown in static (*n* = 3). One-way Anova was performed and * *p* < 0.05, ** *p* < 0.01, *** *p* < 0.001, **** *p* < 0.0001 were considered significant. (**E**) Immunohistochemistry analysis of LGR6 and WNT4 in human FTE tissues grown in dynamic conditions.

**Figure 5 cancers-15-02062-f005:**
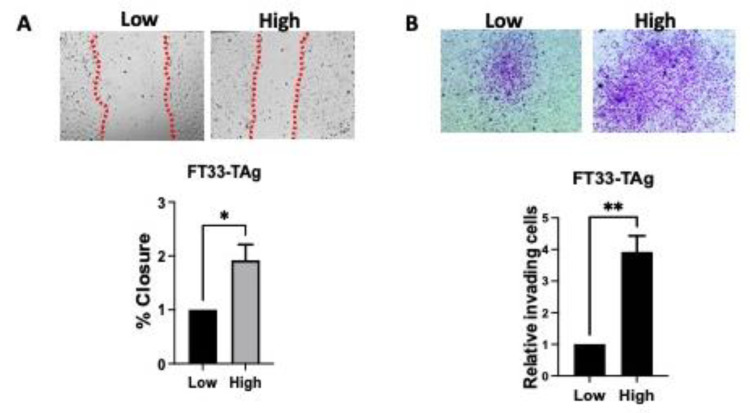
**Effect of testosterone on migration and invasion of FTE cells in static conditions.** (**A**) Wound healing or scratch assay showing increased closure of the gap in the presence of high testosterone indicating increased migration of human FTE cells. (**B**) Boyden chamber invasion assay showing increased invading cells upon increased testosterone. Three independent experiments were conducted for (**A**,**B**) and statistical analysis was done using T-test. * *p* < 0.05 and ** *p* < 0.01, were considered significant.

**Figure 6 cancers-15-02062-f006:**
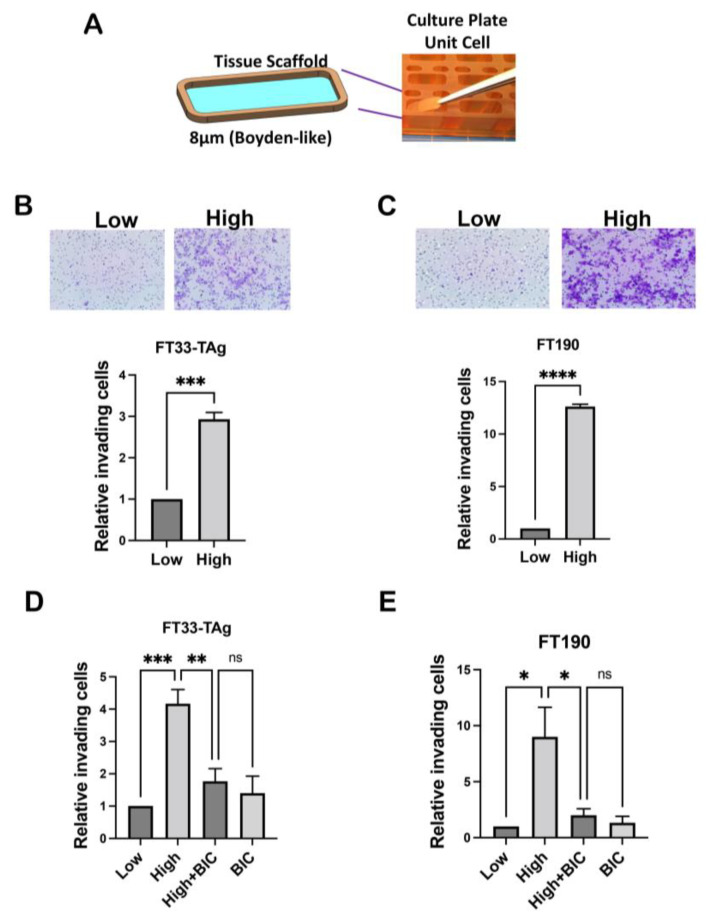
**Effect of Testosterone on FTE cells invasion in the PREDICT-MOS microfluidic system.** (**A**) Schematic showing the insert developed by the Draper laboratory to use for a novel dynamic invasion assay. (**B**,**C**) FT33-TAg or FT190 were plated on Matrigel-coated inserts on the PREDICT-MOS system and exposed to testosterone for 48 hours. Invading cells were counted in 3 independent experiments. T-test was performed and *** *p* < 0.001, **** *p* < 0.0001 were considered significant. (**D**,**E**) Invasion assay on the PREDICT-MOS system for FT33-TAg or FT190 as in previous panels but in the presence of bicalutamide (BIC). One-way analysis was performed and * *p* < 0.05, ** *p* < 0.01, *** *p* < 0.001, were considered significant; ns indicates non-significant.

**Figure 7 cancers-15-02062-f007:**
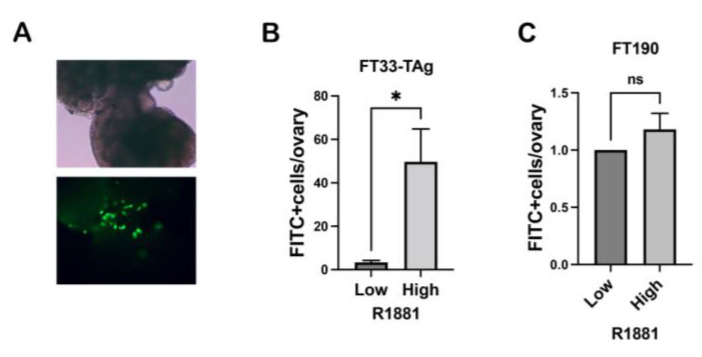
**Role of testosterone on adhesion to ovarian stroma.** (**A**) Bright field (BF) image of the ovary (top) and a fluorescence picture with FTE cells in green (FITC) attached to ovarian stroma (bottom) using 10× objective. (**B**,**C**) Fluorescent FTE cell lines FT33-TAg and FT190 attached to the ovary where counted. Quantification of 3 independent experiments per line. T-test was performed and * *p* < 0.05, was considered significant; ns indicates non-significant.

## Data Availability

For RNAseq experiments RNA was isolated using RNAeasy kit from Qiagen and submitted to the Northwestern NUseq Core and submitted to GEO.

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
