# Peer review of "Increased Local Testosterone Levels Alter Human Fallopian Tube mRNA Profile and Signaling"

_cancers, 2023, doi:10.3390/cancers15072062_

Round 1

Reviewer 1 Report

The authors present a compelling study on the role of testosterone in negatively regulating fallopian tube cells and directing them to an early precursor of high grade serous ovarian cancers. This is the continuation of their previously published work (Human Reproduction, Vol.35, No.9, pp. 2086–2096, 2020) that demonstrated impact of testosterone exposure to the ciliary cells in the fallopian tubes. In this report, they test the hypothesis that testosterone increases the migration and invasion of fallopian tube secretory cells and increases their stem-like cell potential.  They have utilized a variety of models that include: primary human fallopian tube tissues grown ex vivo for 14 days and the immortalized fallopian tube secretory cell lines (FT33, MOE), as well as the Solo-MFP and PREDICT -MOS microfluidic devices.

Overall, their manuscript is well-written, data are well presented, and discussion is easy to follow. Great job on this investigation into the pathologic link between common gynecologic disease (PCOS) and HGSOC initiation, from the lens of androgen signaling!

Major comments:

Can authors comment on how long is the time between presentation of PCOS symptoms and development of SCOUT or STIC? Are PCOS women more likely to be diagnosed with HGSOC that presents with STIC lesions?

What were the rationale for the concentrations of estrogen and testosterone utilized for primary fallopian tube tissue maintenance ex vivo in ‘low’ and ‘high’ testosterone conditions for 14 days? 

The testosterone in ‘low’ condition changes from 0.8nm (day 1-13) to 1.25nM on day 13-14. Meanwhile, it is fixed at 2nM for ‘high’ testosterone condition. Since these concentrations of testosterone are not significantly different from each other (i.e. in order of magnitude), what are the functional difference the authors hypothesize to probe in the resulting experiments?

Solo-MFP microfluidic device RNA-seq data on primary human fallopian tube tissues, appears without any context in results described for Figs. 1, 2 and 3. Can the authors add the description of this device and associated experiments to the introduction section to orient the readers?

Why was Solo-MFP not utilized for the experiments represented in Figs. 4 and beyond?

Are data in Figs. 1A, 1C and 2A collected only from the secretory cells of the ex vivo cultured fallopian tube tissues in low and high testosterone conditions? If yes, how were the secretory cells separated from and other cells (such as ciliated cells) prior to RNA-seq? If not, how are these data different from Fig. 4A, C and Fig. S1 of the authors’ 2020 Human Reproduction paper? If no separation (secretory vs ciliated) of cells from fallopian tube tissues was performed prior to RNA-seq, were the primary PCOS samples from women same between this work and 2020 Human Reproduction paper?

Can the authors add the details of the microfluidic devices, i.e, Solo-MFP and PREDICT-MOS in the legends of the Figures in order to help the readers understand the experimental set up? This detail can include the device type and flow rates.

The authors can move Fig. 5 (PREDICT-MOS device schematic) to the Supplementary Data.

For the data shown in Fig. 4 A and B, which primary fallopian tube tissues were used? Were the same as the ones mentioned in Figs. 1 and 2?

Are the data from Fig. 7 derived from PREDICT-MOS? The description in lines 259-263 and 238-243, makes it hard to understand if it is so.

Were the experiments described in Fig. 8 (lines 308-313) conducted under static or dynamic conditions?

Minor comments:

Line 67: What is SCOUT? Please write the full name before abbreviating.

Line 69: What do authors mean by the following sentence? ‘Mutation and loss of p53 prevent cilia cell lineage..’.

Line 160: Write 10^4 as a subscript.

Lines 205-209: Legend for Figure 1 needs to be corrected to remove one of the captions for  1C, since there are two of them present.

Lines 230-231: There is a typo at the end of the caption for Fig. 3.

Line 266: The authors forgot to add the citation to Cancers 2021, 13(8), 1925.

Line 274-275: One word is missing from the end of the sentence referring to Fig. 7A (ovarian secreted…).

Lines 295-299: In Fig. 7 B, C, D, and E, what does the y-axis represent, and how was it quantified and normalized?

Add information about R1881 inhibitor in the lines 168-173 in the methods.

Line 148: What are MTEC PTEN shRNA or SCR shRNA  cells? Where is the data regarding them described in the manuscript?

Author Response

Thank you for the thorough revision of the manuscript. We have addressed all your comments at the best of our knowledge. The point-by-point response is attached below.

Reviewer 2 Report

This paper is clearly written and well organized. The introduction and background are reasonable given the premise of the paper. The authors explain very well in this paper that increased local testosterone levels alters human fallopian tube genetic profile and signaling. I am recommended to publish this work without any modification.

Author Response

Thank you so much for approving our manuscript.

Round 2

Reviewer 1 Report

The revised manuscript is easier to read and comprehend than the previous version. The changes the authors made are much appreciated!

The following points have not been answered appropriately in requested detail by the authors. 

Can you please respond to the following remaining questions?

The authors have provided a rationale for the value of ‘high’ concentration of testosterone in PCOS. However, no rationale is provided for choosing the ‘low’ concentration value. The value of testosterone and estrogen used in the reference 33, as suggested by authors is: testosterone (10nM – 100mM) and Estradiol (50nM) for ‘maximal physiologic concentrations’. Can the authors provide the definitive references for their choice of both hormones in ‘low’ and ‘high’ conditions?

Please add the flow rates for both Solo-MFP and PREDICT-MOS devices in the new Supplementary Figure S3.

Can the changes observed between testosterone ‘low’ and ‘high’ experimental groups be explained due to potential gradients in nutrients, growth factors, or the hormones (estrogen, testosterone) in the static versus dynamic microfluidic devices? If not, why?

The reference ‘Jackson-Bey, Human Reproduction’ has not been included in the revised manuscript.

Author Response

Comments and Suggestions for Authors

The revised manuscript is easier to read and comprehend than the previous version. The changes the authors made are much appreciated!

The following points have not been answered appropriately in requested detail by the authors. 

Can you please respond to the following remaining questions?

The authors have provided a rationale for the value of ‘high’ concentration of testosterone in PCOS. However, no rationale is provided for choosing the ‘low’ concentration value. The value of testosterone and estrogen used in the reference 33, as suggested by authors is: testosterone (10nM – 100mM) and Estradiol (50nM) for ‘maximal physiologic concentrations’. Can the authors provide the definitive references for their choice of both hormones in ‘low’ and ‘high’ conditions?

Response: We apologize for the missing references. We based our concentration choice on the following references, which have now been added to the paper. Braunstein et al., 2011. J Sex Med; Waldstreicher et al., 1988. J Clin Endocrinol Metab. Line 208.

Please add the flow rates for both Solo-MFP and PREDICT-MOS devices in the new Supplementary Figure S3.

Response: We added the flow rate of 40ul/hour in the figure.

Can the changes observed between testosterone ‘low’ and ‘high’ experimental groups be explained due to potential gradients in nutrients, growth factors, or the hormones (estrogen, testosterone) in the static versus dynamic microfluidic devices? If not, why?

Response: In figure 3 and 4 static and microfluidics show same pattern of gene regulation. However, the small differences could be due to nutrients gradients changed by the flow.

The reference ‘Jackson-Bey, Human Reproduction’ has not been included in the revised manuscript.

Response: This reference was at line 195, however we added it also to line 225.
